# Zika Virus and the Risk of Developing Microcephaly in Infants: A Systematic Review

**DOI:** 10.3390/ijerph17113806

**Published:** 2020-05-27

**Authors:** Evangelia Antoniou, Eirini Orovou, Angeliki Sarella, Maria Iliadou, Nikolaos Rigas, Ermioni Palaska, Georgios Iatrakis, Maria Dagla

**Affiliations:** 1Department of Midwifery, University of West Attica, 12243 Athens, Greece; eorovou@uniwa.gr (E.O.); asare@uniwa.gr (A.S.); miliad@uniwa.gr (M.I.); epalaska@uniwa.gr (E.P.); giatrakis@uniwa.gr (G.I.); mariadagla@uniwa.gr (M.D.); 2Department of Health Management, Greek Open University, 26335 Patra, Greece; nikosstrigas@gmail.com

**Keywords:** Zika virus, Zika virus and pregnancy, Zika virus and microcephaly, Zika virus and congenital malformations

## Abstract

The global epidemic of Zika virus has been a major public health problem affecting pregnant women and their infants. Zika virus causes a viral disease transmitted to humans mainly by the infected *Aedes* mosquito bite. The infection is not severe in most cases; however, there is evidence that infection during pregnancy may be associated with fetal genetic abnormalities (including microcephaly). In addition to microcephaly and other malformations, some specific lesions in the central nervous system have been reported. The aim of this systematic review was to determine the risk of developing microcephaly in infants whose mothers were infected with Zika virus in pregnancy. Epidemiological studies and case reports were incorporated in our review, finally including 15 articles from an initial pool of 355 related papers. Most studies have linked maternal infection during pregnancy to the development of neonatal microcephaly. The period considered most dangerous is the first trimester and the beginning or the whole of the second trimester. In order to understand the relationship between Zika virus and microcephaly in infants, a cohort study will be able to estimate the time from the onset of Zika infection and the full spectrum of adverse pregnancy outcomes.

## 1. Introduction

The purpose of the present study was to analyze all publicly available data to promote an informed discussion of trends in microcephaly cases in infants after mothers’ infection with Zika virus during pregnancy, in Brazil mainly and elsewhere. The study approached the issue after the end of the pandemic.

After Zika virus was first identified in 1947, from the blood of rhesus monkeys in the Zika forest in Uganda, there were few cases reported in humans. In 1952, the first human cases of Zika were detected and since then, outbreaks of Zika have been reported in tropical Africa, Southeast Asia, the Pacific Islands, and Southern America. Before 2007, at least 14 cases of Zika had been documented [1]. However, in 2015, the first case of Zika virus was diagnosed in Southern America linked to cases of microcephaly in infants with malformations and neurological disorders [2].

The Centers for Disease Control and Prevention (CDC) defines the Zika virus as a flavivirus transmitted in humans by *Aedes aegypti* mosquitoes [3] and infection in early pregnancy is linked with microcephaly and other malformations such as damage to the central nervous system and severe developmental disabilities in children [4]. Microcephaly is defined as a head circumference measure that is smaller than a certain measure for infants of the same age and sex. The measure value for microcephaly is more than 2 standard deviations (SDs) below the average. During routine ultrasounds in pregnancy, microcephaly can be diagnosed in the second or early third trimester [5]. A higher prevalence of the Zika virus infection was observed in poor communities, where there is a deficit in prevention and supporting services.

Since 2015, various studies have been conducted, such as reviews and case reports with contradictory results on important points. According to Ellington et al. in 2016, the probability of microcephaly in infants was about 1% to 13%, with limited data for the second and third trimesters [6]. However, other researchers, e.g., Alvarado 2017, claim that the relationship of the microcephaly and Zika virus is in doubt and perhaps there is some connection, but it is not yet clarified [7]. Other investigations show a stronger correlation between Zika virus infections and congenital malformations and even make reference to the presence of endometrial fetal death [8,9]. In light of the findings of an association between Zika virus and central nervous system (CNS) abnormalities, recent studies have shown a strong relation but only the exposed infants with microcephaly or other malformations are more likely to have low cognitive development [10].

### 1.1. Zika Virus Transmission

Zika virus is transmitted from *Aedes aegypti* and *Aedes albopictus* mosquitoes, which are the species more commonly found at 2000 m above sea level. Zika virus is an arthropod-borne virus that is a member of *Flavivirus*, *Pegivirus*, and *Pestivirus*. Flaviviruses include the Zika virus, yellow fever virus, West Nile virus, and dengue virus, while *Hepacivirus* comprises the hepatitis C virus [11].

Zika virus is transmitted in various ways. The *Aedes* mosquito’s bite is the most common one. Transmission from the infected mother to the fetus through the placenta has also been reported. Additionally, Zika virus has the ability to be transmitted through sexual contact; in 2016 the first male-to-male transmission was reported in Texas, United States of America USA). However, the mosquito bite is the most serious threat of transmission [12]. 

According to the CDC, primate mammals are the main “tanks” of the virus, while human-to-human (mosquito-borne) transmission occurs during the viral outbreak. This means that an infected mosquito, can infect a second mosquito not only directly but also through the blood of a human: the first infected mosquito bites the human, transmits the virus, and then a second healthy mosquito bites the same human and gets the virus through the blood. The cycle starts when a mosquito bites an infected person. After a 10 day incubation period, the mosquito’s saliva becomes infected and from that moment a mosquito becomes a vector able to infect a human [13].

### 1.2. Zika Virus Clinical Symptoms and Diagnosis

Zika virus infection in acute stage is believed to be asymptomatic in up to 80% of the infected people and it is classically characterized by low fever, arthralgia, maculopapular rash accompanied by pruritis, and conjunctivitis. After a 12 day incubation period, symptoms usually last only a few days. Apart from the consequences in pregnancy, infections in healthy adults were associated with Guillain–Barre syndrome as well [14]. However, infection during pregnancy can cause intrauterine growth restriction, birth defects, vision and hearing loss, resulting in cognitive and speaking problems accompanied by social and motor development problems in children. Zika virus disease is very often misdiagnosed because the symptoms are similar to the ones of dengue fever and chikungunya.

The diagnostic value of Zika virus disease differs between countries and specific population groups. Travelers in high-risk areas or people in contact with an infected person must be screened with a serological blood test. Furthermore, pregnant women with possible Zika virus exposure, with or without symptoms and women who were diagnosed with fetal microcephaly must be tested for Zika virus infection. Exposed neonates should be evaluated with special Zika Outcomes and Development in Infants and Children (ZODIAC) tools. The current protocol of Zika infection management involves only symptomatic care. Due to the serious problems faced by children exposed during pregnancy and their adverse effects on the economy and society, the early diagnosis of exposure is very important [15,16].

### 1.3. Zika Virus in Europe and Greece

According to the World Health Organization (WHO), there is a Zika virus transmission risk in Europe and this varies from country to country. Thus, on 2 June 2016, the ongoing transmission of Zika virus by mosquitoes was reported from 60 countries/regions worldwide [17]. In Europe, from January 2016 to 26 May 2016, the European Center for Disease Control (ECDC) recorded 638 imported cases in 18 European countries (36 involving pregnant women), while no native mosquito transmission was recorded. In 2018, in European countries, there were 51 reported non-autochthonous vector-borne cases. A percentage of 56.8% of the virus infection in Europe came from the Caribbean, South East Asia, and American travelers. In 2019, a local vector-borne transmission of Zika virus was reported in France. This is the first time that vector-borne transmission of Zika virus through *Aedes albopictus* was reported in Europe [18]. In Greece, the *Aedes aegypti* mosquito has not been recorded for many decades (since the early 1950s); however, in 2003–2004 the *Aedes albopictus* mosquito was recorded in the country and from then on it was recorded in many parts of the country. In Greece, there is also the risk of the introduction of Zika virus mainly by infected travelers from countries with active virus transmission (or less likely by migrants or by infected mosquitoes, which can be transported by public transport or through commercial activities) [19].

## 2. Method

The objective was to systematically review the available evidence that causally links Zika virus and neonatal microcephaly. We conducted a review of different outbreaks of Zika virus disease, including investigations and case reports studies. The research was carried out based on PubMed, Crossref, and Google Scholar. The keywords used were: “Zika virus and microcephaly”, “Zika virus and fetal toxicity”, “Zika virus and craniostenosis”, “Zika virus and congenital malformations”, “Zika virus and teratogenesis”, “Zika virus and children’s mental health”. The timeline was set from February 2016 to December 2019 and out of 355 studies, only 15 were included in the review. More specifically, the articles identified through the initial search strategy were first screened by abstract and title. The full texts of appropriate studies were examined against the inclusion and exclusion criteria and from a total of 338 studies, a total of 150 reviews, systematic reviews, and meta-analyses were rejected. After further screening, 173 articles including other genetic disorders related to Zika, i.e., hearing problems, eye problems, cerebellum problems, hydrocephalus, intrauterine growth restriction, or even the Guillain–Barre syndrome, were rejected too (Figure 1). The study also analyzed the exposure to Zika virus infection after doing appropriate tests (molecular and serology) in blood serum (mother–newborn), in amniotic fluid, in urine, and in the mother’s suspect Zika virus symptoms (low fever, rash, conjunctivitis, muscle and joint pain, or headache), which were confirmed by examinations. As an outcome, we analyzed the frequency of microcephaly in infants (smaller head circumference more than 2 SDs depending on the sex and age). All infants with microcephaly underwent neural imaging examinations. Regarding the methodological quality of the articles, nine criteria were used to rate them. The first criterion, concerning the representative exposure sample was met by all studies except the two clinical ones. Regarding the second criterion, only four studies were graded because the other studies concerned infants or mothers who had been exposed to Zika virus. All studies met the third criterion because the exposure was identified with laboratory tests. In all studies except de Araújo et al. [20], the outcome did not precede the study. The fifth criterion, which was the adaptation for the educational level, did not exist in any study. In all studies apart from three, an adaption for an additional confounding factor had been used. All studies met the seventh criterion for evaluating the effects of microcephaly through examinations and also, in all studies there was sufficient follow-up time, thus fulfilling the eighth criterion. Finally, all studies met non-bias of wear, the ninth criterion. The score of studies varied between 8 and 6. No research had a score of 9 because all studies had a negative response to criterion5 (Table 1).

## 3. Results

The 15 articles included to this systematic review have been carried out in various countries in the territory of Latin America, USA and Europe. Specifically, eight studies were conducted in Brazil, one in Colombia, five in USA, and one in Finland. Of the total studies, six were cross-sectional studies, one case control, two clinical trials with experimental animals, five case studies, and one cohort study (Table 2). According to the methodological evaluation of the studies, 11 studies were very good and 4 were of moderate methodological quality (Table 1).

More detailed, in the study of Magalhães et al., 2016 [21], which was carried out in Brazil, all available data support a causal relationship between the Zika virus and related microcephaly and (CNS) abnormalities in infants. The 8301 cases of microcephaly were analyzed and confirmed by laboratory tests and neuroimaging methods, thus excluding incidents due to other congenital abnormalities. The results showed that 15% of the patients with microcephaly was related to Zika virus infection. According to the Melo et al. 2016 [22] study, two newborns with congenital Zika infection were observed from their intrauterine life. The infection was found during the first twotrimesters of pregnancy and confirmed with maternal–fetal laboratory tests (mother serological tests, amniocentesis). Both neonates were shown brain damage with reduced brain volume, brain celiac disease, cerebral hypoplasia, hydrocephalus, and arthropathy. The cross-sectional study of Sarno et al., 2016 [23], had the aim to describe CNS lesions in fetuses with microcephaly after congenital Zika infection. The researchers concluded that Zika syndrome is linked to neurological abnormalities such as abdominal pain, abdominal calcifications and posterior cingulate damage which become apparent only at the end of the second trimester and beyond. Furthermore, referring to CNS lesions, they observed that damage to the posterior cavity and articulation are unusual findings in other infections and their presence may indicate Zika infection. In addition, the 1501 cases studied by Franca et al., 2016 [24], were the first to be fully investigated and funded by the Brazilian Ministry of Health. The purpose of the research was to identify infants with microcephaly and make a complete clinical record. From this study, 602 cases were finally confirmed as congenital Zika syndrome. The research also shows that the earlier the rash is reported in pregnancy, the smaller the circumference of the newborn’s head.

The case–control research of de Araújo et al., 2016 [20], investigated the relationship between infection with Zika virus and the occurrence of neonatal microcephaly. The cases studied consisted of 30 infants with microcephaly and 62 infants without outcome. Infants selected as controls had the same probable date of birth as patients and similarly with premature infants. The examination was followed by a blood sample from the umbilical cord and cerebrospinal fluid (CSF) of the infants with simultaneous laboratory tests for their mothers. The results of this study showed that 80% of 30 mothers of cases had Zika virus infection compared with 64% of 62 control mothers (*p* = 0.12).

There were three case studies in our review. In the case report of Driggers et al., 2016 [25], a woman in the 11th week of her pregnancy reported mosquito bites after a vacation with her husband in Mexico. The day after their arrival in Washington, the woman and her husband experienced symptoms such as low-grade fever, ocular pain, and myalgia for five days. Initially, the results showed positive IgG and negative IgM against dengue virus, but subsequent serological analysis showed positive IgG–IgM against Zika virus, while serological analysis for chikungunya virus was negative. Furthermore, at 16 weeks, the presence of the virus was detected by RT-PCR, thus confirming infection with Zika virus. After the termination of pregnancy with the consent of the mother at 21 weeks of gestation, the brain had the highest viral load, weighed 30 g (reference weight, 49 ± 15 g) and neuropathological features of Zika infection. In a case study in 2016, Sarno et al. 2016 [26], reported the case of a 20-year-old pregnant woman following a major outbreak of Zika virus Brazil. Following the standard ultrasound examination at week 18, low birth weight was observed for the week of gestation. Ultrasound scans at weeks 26 and 30 showed microcephaly; however, at week 32, an ultrasound showed no fetal heartbeat. Subsequently, Zika virus was detected in samples from the placenta, brain, CSF, and viscera. Moreover, in Martinez et al., 2016 [27], samples from two neonates with microcephaly (estimates gestational ages of 36 and 38 weeks), dying within 20 hours of birth and two embryos miscarried at 11 and 13 weeks were sent to the CDC by the state of Rio Grande do Norte in Brazil for the evaluation of histopathological and laboratory tests after high suspicion of Zika infection. The four mothers had developed symptoms of the Zika virus infection, including fever and rashes during the first trimester of pregnancy, but they had no clinical symptoms of active infection during labor or miscarriage.

In the cross-sectional study of Pacheco et al., 2016 [28], data on 11,944 cases of pregnant women were collected from the Colombian National Public Health Surveillance System. In the cases of symptoms reporting, we see that from the reported start date and after serum sample collection for RT-PCR viremia can last up to 21 days. In another case, the virus was detected in the blood serum at 4 and 10 weeks, but not after childbirth, which indicates a persistence of viremia in pregnancy. In the Calvet et al., 2016 [29], case study with data from the Brazilian National Public Health Surveillance System, two cases of embryos with microcephaly were studied. Amniotic fluid samples were examined by RT-PCR, and the Zika virus genome was identified in two pregnant women, while at the same time tests for dengue virus, chikungunya virus, *Toxoplasma*, rubella, herpes simplex virus, HIV, *Treponema pallidum*, and B19 parvovirus were all negative.

During the period October 2015–February 2016, several cases of fetuses with microcephaly were reported to the Paraiba Institute of Embryonic Medicine, Brazil. Eleven of the 150 pregnant women were selected for this particular study of Melo et al., 2016 [30], due to findings of brain damage during fetal ultrasound imaging. Amniocentesis was performed on all women for laboratory confirmation of Zika virus infection by (PCR). After an initial evaluation of the ultrasound indications, periodic follow-up was performed by fetal medicine specialists as well as magnetic resonance imaging MRI of fetuses; however, 3 of the 11 newborns died shortly after birth. This study analyzes the course from pregnancy to childbirth of 11 embryos exposed to endometrial Zika infection, presenting with prenatal and postnatal brain damage as well as other developmental abnormalities. Moreover, despite the fact that most infants had microcephaly measuring the circumference of the skull, some infants had a measurement that was consistent with their gestational age, as brain atrophy was offset by the circumference of the skull.

In addition, a study by Rice et al., 2018 [31], performed a follow-up on 1450 children (over 1 year) of mothers who had been infected with the virus during pregnancy. From the existing sample the 14% had birth defects associated with Zika virus syndrome and 6% of them had microcephaly. In the Hoen et al., 2016 [32] study, 527 infants from mothers with laboratory confirmation of Zika virus were included. From this research, the risk of microcephaly is higher in the first trimester of pregnancy infection than in the second while increased rates of microcephaly are also present when the infection occurs in the third trimester. Therefore, according to the findings the increased rates of microcephaly (moderate or proportionate) in the third trimester show the increased risk of microcephaly throughout pregnancy.

Finally, Miner J et al., 2016 [33], in an experimental study of mice in the United States from January to February 2016, carried out tests to determine the degree of association of Zika virus with neonatal microcephaly as well as with other congenital disorders. Scientists, referring to the urgent need to investigate intrauterine Zika transmission and its pathological consequences, developed two models of Zika virus infection during pregnancy. In the first model, Zika crosses the placenta barrier resulting in fetal infection and CNS damage and residual intrauterine growth, in more severe cases the infection resulted in fetal death, in the second case, after treatment with anti-interferon (IFN)-AB antibodies in the placenta, infection of the developing embryo occurred but was less severe and did not cause the death of the fetus. Furthermore, in Noguchi et al.’s 2019 [34] study, the effect of the Zika virus infection on the brain and the CNS of the mouse are presented—resulting in massive neurodegeneration of infected regions. The Brazilian Zika virus stain is also reported in the research and produces particularly devastating neuropathology in the fetal brain opposed to the viral French Polynesia stain.

## 4. Discussion

The present study provides strong evidence for a probable link between Zika virus infection and congenital malformations resulting in serious malformations in children. We see a clear lead in studies from Brazil and Latin America in general that the virus is in the form of an epidemic and the lack of prospective studies so far are due to the short time of the virus and microcephaly worldwide.

The results, according to the articles, show that the prevalence of microcephaly cases in Southern America (confirmed by laboratory tests), has been shown to be related to maternal infection with Zika virus in pregnancy, especially in the first and second trimesters [25]. Particularly in the first trimester, the disorder is associated with the process of neuronal migration and spinal cord formation. Of course, while the first trimester appears to be more prone to risk than the second trimester, cases of microcephaly are confirmed (or not) by reporting mothers’ infection symptoms at weeks 18–20 [30]. The fact that 80% of infections in pregnant women are asymptomatic, while high viral load values in the fetal brain, placenta, and fetal membranes are isolated, indicates that the timing of infection is crucial for the outcome of the disease [25]. Therefore, the risk of developing Zika virus infection is greater in the first and second trimesters of pregnancy as the CNS and brain develop. It is also of importance that Zika does not cross only the placental barrier. Elements of the virus genome were found in breast milk, saliva, urine, and mother’s blood serum even a few days after childbirth [28,29].

Certainly, the presence of these brain lesions is not necessary for the diagnosis of microcephaly. Some infants with the outcome did not show any brain damage during the imaging procedures presented. This suggests that congenital Zika syndrome may occur in infants with microcephaly and without the presence of imaging brain abnormalities. According to these findings, public health surveillance authorities should not disregard patients with microcephaly without findings from imaging methods [5]. Concerning the prevention of virus infection, so far all epidemiological studies have shown that Zika infection during pregnancy causes catastrophic neurodevelopmental effects on human embryos, but there is currently no effective treatment or prevention of Zika virus infection due to exposure to *Aedes* mosquitoes. Inoculation attempts in experimental animals before pregnancy, early, or mid-pregnancy, indicate fetal death or genetic abnormalities with microcephaly [31].

Regarding preventive measures, international organizations (CDC/MEDICHEM/WHO) have made recommendations on Zika virus and microcephaly in infants. The CDC recommends measures to prevent disease and transmission. The best way to prevent is a protection with special mosquito repellent measures and the avoidance of direct contact of people with Zika virus, due to patient’s body fluids as the infection is transmitted in this way [35]. Prevention in pregnancy involves appropriate family planning methods in the endemic areas, thorough parental care in affected countries, avoiding unnecessary trips to endemic countries, especially by people of reproductive age. Based on the available data, a woman who is not pregnant will not be at risk for birth defects in future pregnancies, as long as relative immunity has developed. According to the above recommendations, the importance of prevention by avoiding travel to endemic countries by people of both sexes is emphasized. Contraceptive measures in confirmed cases are proposed at least for a period of 6 months after infection in men. Currently there is no cure or vaccine to prevent Zika virus, the treatment is coincidental and the only way to prevent it is to prevent mosquitoes breeding in stagnant waters, insecticides, and bite protections [36].

The World Health Organization (WHO), in response to the outbreak of Zika virus, has set out a response plan, outlining four key goals with emphasis on the prevention and management of neonatal microcephaly caused by Zika infection. These goals include detection, prevention, care support, and research. Given the current situation, the spread of Zika virus is expected to have long-term effects on health, for families, communities, and countries whose health systems will be called upon to care for children born with these complications for years to come [37]. Finally, giving the limitations of cross-sectional studies (that do not account for the incidence of the disease) and case studies (that provide a minimum basis for scientific generalization), a design of cohort studies is the most appropriate to established the relationship between Zika virus and microcephaly in infants.

## 5. Conclusions

In the present study, an attempt was made to approximate the frequency and conditions under which Zika may cause microcephaly as well as factors increasing the risk of vertical transmission in pregnancy.

Most studies have linked maternal infection during pregnancy to the appearance of brain damage in newborns. Consequently, congenital microcephaly is the hallmark of Zika’s intrauterine infection. The period considered most dangerous for the vertical transmission of the infection is the first trimester and the beginning of the second or the entire second trimester, as with the Toxoplasmosis, Rubella, Cytomegalovirus and Herpes TORCH and all pathogens.

Therefore, for countries with outbreaks of Zika, it is advisable to prevent unintended pregnancies and take measures to combat transmission of Zika virus from infected mosquitoes, especially during the first and second trimesters of the nervous system development.

In order to better understand the relationship between Zika infection and neonatal microcephaly, further research will be needed, because the prevalence of Zika infection and risk of vertical transmission remains high. A cohort study of pregnant women will be able to estimate the time from onset of Zika infection and associate it with the full range of adverse pregnancy outcomes.

Zika virus epidemics have significant short and long-term impacts in the socioeconomic spheres in the Americas. In addition to the huge financial losses due to a heavy dependence on tourism and increased stress on the healthcare system, affected individuals with microcephaly may be unable to join the labor force in the future. The Zika epidemic has disproportionately affected the poorest countries, as well as the vulnerable groups creating unequal social and health service and contributing to widening inequalities in the region. As recently observed, epidemics spread by mosquitoes (Zika, yellow fever, dengue, chikungunya) can expand rapidly, including to other parts, which highlights the need for effective control of the vectors.

## Figures and Tables

**Figure 1 ijerph-17-03806-f001:**
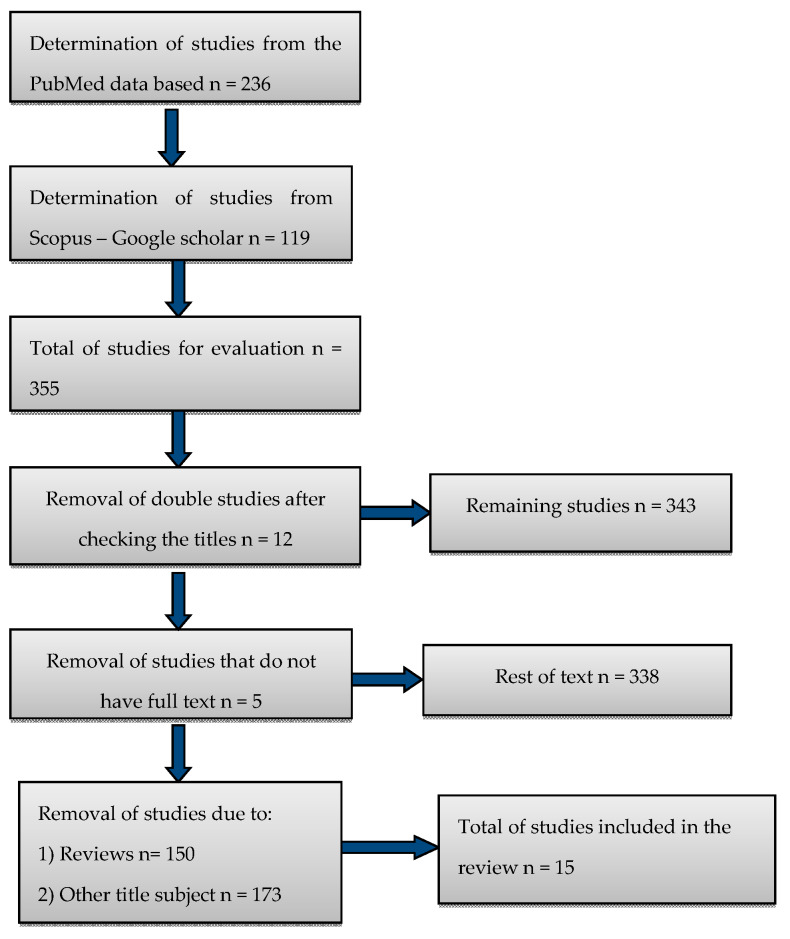
Chart.

**Table 1 ijerph-17-03806-t001:** Evaluation ofsurveying methodological quality studies.

Author/Year	Selection 1 2 3 4	Comparability 5 6	Result 7 8 9	TOTAL
1	* - * -	- *	* * *	7
de Araújo (2016) [20]				
2	* - * *	- *	* * *	7
Magalhães (2016) [21]				
3	* - * *	- *	* * *	7
Melo (2016) [22]				
4	* -* *	- *	* * *	7
Sarmo (2016) [23]				
5	* * * *	- *	* * *	8
Franca (2016) [24]				
6	* -* *	- *	* * *	7
Driggers (2016) [25]				
7	* - * *	- *	* * *	7
Sarno (2016) [26]				
8	* -* *	- *	* * *	7
Martines (2016) [27]				
9	* - * *	- *	* * *	7
Pacheco (2016) [28]				
10	* - * *	- *	* * *	7
Calvet (2016) [29]				
11	* - * *	- -	* * *	6
Melo (2016) [30]				
12	* * * *	- *	* * *	8
Rice (2018) [31]				
13	* * * *	- *	* * *	8
Hoen (2018) [32]				
14	* - * *	- *	* * *	7
Miner Jon (2016) [33]				
15	- * * *	- -	* * *	6
Noguchi (2019) [34]				

Notes: 1. Representative exposure sample, 2. selection of non-exposed, 3. exposure finding, 4. outcome did not precede the study, 5. adaptation for educational level, 6. adaptation for additional confounding factor, 7. outcome evaluation, 8. adequate monitoring time, 9. non-bias of wear. The symbol (*) means that the study met the specific criterion and the symbol (-) means that the study did not meet it.

**Table 2 ijerph-17-03806-t002:** Included in the review.

	Autors/Year	Design	Start Expiry	N	Data	Population	County	Outcome
1.	de Araúj (2016) [20]	Case-Control	January 2016 May 2016	948	Hospitals	Infants	Brazil	Microcephaly
2.	Magalhães (2016) [21]	Cross- Sectional Study	January2016 July 2016	8301	Health Servises	Infants	Brazil	Microcephaly
3.	Melo (2016) [22]	Case Study	October 2015 February 2016	2	Research Institute	Pregnant women-Infants	Brazil	Microcephaly
4.	Sarmo (2016) [23]	Cross-Sectional	July 2015 February 2016	60	University Hospitals	Fetusses	Brazil	Microcephaly
5.	Franca (2016) [24]	Cross- Sectional	November 2015 February 2016	1501	Hospitals	Infants	Brazil	Microcephaly
6.	Driggers (2016) [25]	Case-Study	November 2015 March 2016	1	Hospital	Adults–Infants	Finland	Microcephaly
7.	Sarno (2016) [26]	Case-Study	July 2015 January 2016	1	Pediatric Hospitals	Adults–Infants	USA	Microcephaly
8.	Martinez (2016) [27]	Case-Study	May 2015 February 2016	11	Research Institute	Fetuses–Infants	Brazil	Microcephaly
9.	Pacheco (2016) [28]	Cross Sectional	August2015 April 2016	11,944	Public Hospitals	Adults–Infants	Columbia	Microcephaly
10	Calvet (2016) [29]	Case-Study	February 2016 June 2016	2	Public Hospitals	Adult–Infants	Brazil	Microcephaly
11.	Melo (2016) [30]	Case-Study	October 2015 February 2016	11	National Research	Fetuses	Brazil	Microcephaly
12.	Rice (2018) [31]	Cross Sectional	February 2017 February 2018	1450	PediatricHospitals	Children	USA	Microcephaly
13.	Hoen (2016) [32]	Cohort Study	March 2016 November2016	527	Research Institute	Pregnant Women–Infants	USA	Microcephaly
14.	Miner J (2016) [33]	Clinical Study	January 2016 February 2016	-	University Washington	Mice	USA	Microcephaly
15.	Noguchi (2019) [34]	Clinical Study	2019	-	Research Institute	Mice	USA	Microcephaly

Notes: The cases of microcephaly were defined as moderate (the head circumference was between 2and 3 standard deviations (SDs) for gestation age) and severe (the head circumference was more than 3SDs for gestation age). Moderate microcephaly defined as proportionate (the infant was small for gestation age) and disproportionate (the infant was not small for gestation age).

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
