# Peer review of "Zika Virus and the Risk of Developing Microcephaly in Infants: A Systematic Review"

_ijerph, 2020, doi:10.3390/ijerph17113806_

Round 1

Reviewer 1 Report

1. why time line is only 2016 to 2019? is there any specific reason or logic for this choosing of time line!

2. total study count is 358 while you have written 355? clarify this mistake!

3. Are studies or review area or region specific? if not than only 15 studies globally create doubt on your search strategy and inclusion and exclusion criteria?

Author Response

Dear Reviewer 1

Thank you very much for your review of my article!

  1. why time line is only 2016 to 2019? is there any specific reason or logic for this choosing of time line!(The studies on zika virus and its impact on neonates started in 2016, immediately after the start of the epidemic.)
  2. total study count is 358 while you have written 355? clarify this mistake!(355 was not a mistake! The problem in line 148 was corrected)
  3. Are studies or review area or region specific? if not than only 15 studies globally create doubt on your search strategy and inclusion and exclusion criteria?(Zika epidemic was mainly observed in specific areas; this is why, the majority of the studies refer to them. Furthermore, a case study carried out in Finland observed that the transmission of the virus was the result of a travel to an zika-epidemic area).

Reviewer 2 Report

A systemic review of Zika virus infection and its relationship to microcephaly is of high value to clinicians, public health officials, and researchers. The authors set out to analyze "all publicly available data" and ended up with only 15 articles out of 355 related papers to review.

It is recommended that "systemic review" be included in the title to facilitate appropriate searches. It is also recommended that the abstract be written in a structured format  to provide clarity and a concise description of the review for readers. In addition to background information on why such a review is warranted, it should minimally include data sources, study selection criteria/eligibility for inclusion, results, conclusions, and limitations. 

Introduction

The introduction needs to highlight why this systemic review is needed. What will it add to what is already known? Is this the first systemic review? 

The first paragraph is somewhat disorganized and I suggest that it could be rewritten  more concisely and bring out why we need this review.The historical information on Zika virus is interesting, but I suggest to reorganize this paragraph and move the information on the mosquito etc. to paragraph 1:1. There are some spelling and grammatical errors corrected as follows: Rhesus (line 33), Pacific Islands (line 35), The Centers for Disease Control and Prevention (line 43 and elsewhere whenever the CDC is mentioned).

Similarly, I would suggest reorganizing paragraph 1.2 e.g. Line 82, "After a 12-day incubation period.... is better placed after the first sentence which describes symptoms ending on line 78. This would help the reader not have to jump around the text. 

The paragraph on the diagnosis (line 84) is very unclear. How does the diagnosis differ in different countries. What is the ZODIAC test? The diagnosis of exposure is very important in evaluating the articles for the systematic review and this needs to be expanded.  

Paragraph 1.3. Please add a reference for the WHO and ECDC statements line 90 and 93 as well as line 96. In line 95, ....51 no autochronous .... should read 51 non-autochthonous vector-borne cases.

Methods

I would like to see the objective clearly stated here. In line 106, the authors state that they conducted a review of different outbreaks of Zika virus disease. Is that accurate or was it a review all articles publicly available and happened to cover outbreaks in different countries. Why was Medline not used as a resource?  Line 111 - "researches" - do the authors mean articles?

Please describe more concisely your criteria for exposure to Zika virus  in reviewing the articles.  Did you have a protocol for assessing exposure in the articles. What were considered "appropriate antibody tests" and "appropriate symptoms"? Were any articles excluded if this was not clear in their text?

The outcome measure (line 114) is stated as "incidence of microcephaly" but do these studies actually measure incidence? Is association, prevalence or frequency more accurate. The definition of microcephaly should be described here in the methods and not in the introduction. Also did you extract information on neural imaging which is reported in your description of some of the studies. Was this universal? The methods are lacking in a description of your data extraction protocol.

I recommend moving the flow chart (fig 1) and the sentence about whittling down the 355 reports to 15 to the Results section.

Results

Please explain the removal of 105 studies due to "revisions" and 173 studies due to "other title subject". 

Line 138: Clinical case reports do not really qualify as epidemiological studies - perhaps just use the word "articles" instead of "epidemiological researches".

The sentence (line 143-145) belongs in the discussion.

Table1:

I suggest using United Sates of America or USA and not "America" .

There is an error in #7 - Sarmo 2016 is a case study and #9 Pacheco 2016 is a cross-sectional study not a case control

The table includes p values - for what?  No statistical relationships are provided. This needs clarification.

In general the results are presented in an inconsistent manner, with  insufficient detail to make sense of some of the studies, and much more detail given to  case reports such as Driggers et al 2016 (lines 176-184) and Martinez 2016, (lines 189-1950). The text does not reveal their special relevance.  It would be helpful to have a more uniform approach to presenting the data, e.g exposure data, timing of exposure in pregnancy, fetal versus neonatal detection of microcephaly, number/percentage with microcephaly, neural imaging, comparison group if any.  

In the Magalhães study (lines 153, 154), does the data in this cross-sectional study suggest rather than support a causal relationship between Zika virus and microcephaly. More details on the statistical findings should be reported. Delete "for confirmation (line 156) as it is repetitive.

The Melo study (lines 157-1610 appears to have followed 11 exposed cases from intra to extrauterine life, suggesting that this is a cohort study of 11 cases. There is no information about the denominator (i.e. 11 out of how many). In line 160, I suggest replacing "have been shown" with "were shown". Later in the results section, it appears that this study is mentioned again in better detail. It seems there are 2 publications by Melo et al of the same study (Ref 20 and 28). Would this not exclude one them from the systematic review as a double study? If they both are distinctive publications reporting on different aspects of the study, I suggest not separating them in the text. 

The sentence about the Sarmo et al 2016 cross-sectional study beginning "the researchers concluded a case........is confusing and needs to be rewritten. Do you mean to say: "The researchers concluded that Zika syndrome is linked to abnormalities such as ...... "  No data is presented to support the Franca 2016 conclusion. More detail is provided for the study design in the Araijo et al 2016 study (lines 170-175), but the findings are not included.

In the Calvet case study (lines 200-205),  please rewrite the phrase:" 2 cases  of pregnant and embryos with microcephaly were studied" This does not make sense. 

I suggest moving the "methodological evaluation of research studies to the methods" describing the 9 criteria, and in the results section describing the results as in the table. Please make the table clearer. It is of interest that only 3 studies had a non-exposure comparison group.  I am confused by criterion #4 - Outcome of the study not available. Please clarify these criteria in the methods.

Discussion and Conclusions: 

I suggest changing "establishes" (line 249) to "provides strong evidence for". Also The results do not show rates but rather frequency or prevalence of microcephaly (line 251). The sentence 254-254-256 beginning "Of course" is confusing as written. Do you mean to say that while exposure during the first trimester carries greater risk of microcephaly, cases at 18-20 weeks  confirmed by the onset of maternal symptoms have been reported. Line 261 - delete "alone" as repetitive.

A reference is needed for the recommendations by the CDC/MEDICHEM/WHO) - line 274.

The discussion does not adequately provide the reader with what is learned from this systemic review other than what is already known.  It also does not discuss the limitations of the cross-sectional and case studies reviewed in establishing causality and the need for population based prospective studies to do so.   It also does not mention the need for more studies of the longterm outcomes especially for those without abnormal neurological imaging or those with proportionate microcephaly. 

Line 302 - There is an error- "prevent and prevent pregnancy"

It would be good to mention the tremendous public health and economic costs of the Zika virus epidemic in all affected countries and then highlight the situation in Greece. 

I do hope you find my suggestions and critiques helpful. You have undertaken a big task in doing this systematic review on such a recent and important global health issue. 

Author Response

Dear reviewer

Thank you very much for your review and your suggestions in my article that helped me a lot. I quote you the changes in detail:

It is recommended that "systemic review" be included in the title to facilitate appropriate searches. (Corrected)

 It is also recommended that the abstract be written in a structured format  to provide clarity and a concise description of the review for readers. In addition to background information on why such a review is warranted, it should minimally include data sources, study selection criteria/eligibility for inclusion, results, conclusions, and limitations.(The abstract is written in a structured format without headings, as set out in the rules of the journal. It includes the purpose of the study, introduction, methodology, results and conclusions).

The introduction needs to highlight why this systemic review is needed. What will it add to what is already known? Is this the first systemic review? (Many studies have been carried out after the 2015 Zika epidemic associating zika and microcephaly in neonates. We are not absolutely sure, though, at least up to now, about the impact of the infection on the Central Nervous System of the fetus, while the problem is still a Public Health concern).

The first paragraph is somewhat disorganized and I suggest that it could be rewritten  more concisely and bring out why we need this review.(Added, line 33-34)

The historical information on Zika virus is interesting, but I suggest reorganizing this paragraph and moving the information on the mosquito etc. to paragraph 1:1.(The paragraph was moved, line 66-69)

 There are some spelling and grammatical errors corrected as follows: Rhesus (line 33), Pacific Island(line 35), The Centers for Disease Control and Prevention (line 43 and elsewhere whenever the CDC is mentioned)( Corrected, lines 35, 38, 46)

Similarly, I would suggest reorganizing paragraph 1.2 e.g. Line 82, "After a 12-day incubation period.... is better placed after the first sentence which describes symptoms ending on line 78. This would help the reader not have to jump around the text (It was reorganized, line 85-87)

The paragraph on the diagnosis (line 84) is very unclear. How does the diagnosis differ in different countries? (Corrected and answered in the text, line 95-97))

WhatistheZODIACtest? (Explained, line 99-100)

The diagnosis of exposure is very important in evaluating the articles for the systematic review and this needs to be expanded.  (Supplemented, line 101-103) Paragraph 1.3. Please add a reference for the WHO (literature was added, line 107)and ECDC statements line 90 and 93) as well as line 96. In line 95,(ECDC literature in line 114) ...51 no autochronous.... should read 51 non-autochthonous vector-borne cases. (Corrected, line 111)

Methods

I would like to see the objective clearly stated here.(Added, line 122-123)

 In line 106, the authors state that they conducted a review of different outbreaks of Zika virus disease. Is that accurate or was it a review all articles publicly available and happened to cover outbreaks in different countries. (Many studies were carried out during the epidemic period, especially in the affected areas. The specific systematic review used the most reliable articles from studies funded by the national Public Health authorities). Why was Medline not used as a resource? (One of the interfaces for searching Medline is PubMed. PubMed is provided by Medline for free access. If we had used both databases, we would have to remove more double copies in the end) Line 111 - "researches" – do the authors mean articles? ( studies, corrected)Please describe more concisely your criteria for exposure to Zika virus  in reviewing the articles.  Did you have a protocol for assessing exposure in the articles. (more analytically described, line 129-135)What were considered "appropriate antibody tests" and "appropriate symptoms"? (the appropriate antibody tests are more analytically described in line 130-131. In terms of the “appropriate symptoms” you most probably refer to the mother’s symptoms that in line 131 are described as “mother’s suspect symptoms”) Were any articles excluded if this was not clear in their text?(The text of all articles used was clear) The outcome measure (line 114) is stated as "incidence of microcephaly" but do these studies actually measure incidence?  (the outcome of all articles was microcephaly)

Is association, prevalence or frequency more accurate. (Yes, more accurate) The definition of microcephaly should be described here in the methods and not in the introduction. (The definition of microcephaly was added, line 133-135)Also did you extract information on neural imaging which is reported in your description of some of the studies. Was this universal? The methods are lacking in a description of your data extraction protocol.(All neonates were subjected to imaging methods, line 135)I recommend moving the flow chart (fig 1) and the sentence about whittling down the 355 reports to 15 to the Results section.( In all systematic reviews, the flow chart is included in methodology).

Results

Please explain the removal of 105 studies due to "revisions" and 173 studies due to "other title subject". (105 studies were rejected due to reviews and systematic reviews. 137 were rejected because of no appropriate subject, i.e. zika virus and Guillain-Barren syndrome …)

Line 138: Clinical case reports do not really qualify as epidemiological studies - perhaps just use the word "articles" instead of "epidemiological researches".(Corrected, line 167)

The sentence (line 143-145) belongs in the discussion.(Moved todiscussion,line 286-289)

Table1:I suggest using United Sates of America or USA and not "America" (Corrected)

There is an error in #7 - Sarmo 2016 is a case study and #9 Pacheco  2016 is a cross-sectional study not a case control (Corrected)

The table includes p values - for what?  No statistical relationships are provided. This needs clarification.(We want to show that the results of each study were statistically significant)

In general the results are presented in an inconsistent manner, with  insufficient detail to make sense of some of the studies, and much more detail given to  case reports such as Driggers et al 2016 (lines 176-184) and Martinez 2016, (lines 189-1950). The text does not reveal their special relevance.  It would be helpful to have a more uniform approach to presenting the data, e.g exposure data, timing of exposure in pregnancy, fetal versus neonatal detection of microcephaly, number/percentage with microcephaly, neural imaging, comparison group if any.  (Extra information added)

In the Magalhães study (lines 153, 154), does the data in this cross-sectional study suggest rather than support a causal relationship between Zika virus and microcephaly. More details on the statistical findings should be reported.(More findings of the research have been added, line 187-188)

Delete "for confirmation (line 156) as it is repetitive(Deleted, line 186)

The Melo study (lines 157-1610 appears to have followed 11 exposed cases from intra to extrauterine life, suggesting that this is a cohort study of 11 cases. There is no information about the denominator (i.e. 11 out of how many). In line 160, I suggest replacing "have been shown" with "were shown". (Corrected,line 191)

Later in the results section, it appears that this study is mentioned again in better detail. It seems there are 2 publications by Melo et al of the same study (Ref 20 and 28). Would this not exclude one them from the systematic review as a double study? If they both are distinctive publications reporting on different aspects of the study, I suggest not separating them in the text. (It was a mistake. Melo’s study concerned was related to the case study and not the cross sectional one. The two studies are different but carried out by the same researcher. They were published in different journals. The mistake was corrected and the correct literature was added).

The sentence about the Sarmo et al 2016 cross-sectional study beginning "the researchers concluded a case........is confusing and needs to be rewritten. Do you mean to say: "The researchers concluded that Zika syndrome is linked to abnormalities such as ...... (The sentence  was corrected, line 195)

"  No data is presented to support the Franca 2016 conclusion. (Research data were added, line 199-203) More detail is provided for the study design in the Araijo et al 2016 study (lines 170-175), but the findings are not included.(Corrections were made, line 205-206 and the findings of the study were more analytically added, line 209-210) In the Calvet case study (lines 200-205),  please rewrite the phrase:" 2 cases  of pregnant and embryos with microcephaly were studied" This does not make sense. (The sentence was corrected, line 237)I suggest moving the "methodological evaluation of research studies to the methods" describing the 9 criteria, and in the results section describing the results as in the table.(The methodology quality table was transferred to the Methods section, line 161-164) Please make the table clearer. (The table was improved; when the article was submitted to the journal, it had this form) It is of interest that only 3 studies had a non-exposure comparison group. I am confused by criterion #4 - Outcome of the study not available. (Corrected, line 162-163) Please clarify these criteria in the methods.(The criteria are reported in the methods section, methods line 136-138)

Discussion and Conclusions: 

I suggest changing "establishes" (line 249) to "provides strong evidence for".(Corrected)

Also the results do not show rates but rather frequency or prevalence of microcephaly (line 251). (Corrected, line 290)

 The sentence 254-254-256 beginning "Of course" is confusing as written. Do you mean to say that while exposure during the first trimester carries greater risk of microcephaly, cases at 18-20 weeks confirmed by the onset of maternal symptoms have been reported. Line 261 - delete "alone" as repetitive.(Corrected, line 303, the risk is bigger in the first and second trimester that the fetus’ central nervous system is developing, although possible infection in the third trimester is not excluded).

Line 261 - delete "alone" as repetitive. (Corrected, line 301)

A reference is needed for the recommendations by the CDC/MEDICHEM/WHO) - line 274.(CDC literature added, line 318, MEDICHEM line 327 and WHO line 333 literature existed)

The discussion does not adequately provide the reader with what is learned from this systemic review other than what is already known.  It also does not discuss the limitations of the cross-sectional and case studies reviewed in establishing causality and the need for population based prospective studies to do so (Line 333-336, the limitations of the studies used were added and the need to develop cohort studies was clarified) It also does not mention the need for more studies of the longterm outcomes especially for those without abnormal neurological imaging or those with proportionate microcephaly.

Line 302 - There is an error- "prevent and prevent pregnancy"(Corrected,  line 345)

It would be good to mention the tremendous public health and economic costs of the Zika virus epidemic in all affected countries and then highlight the situation in Greece(Greece was removed; the significance of the epidemic at a macroeconomic and microeconomic level in the affected countries was stressed, 352-358)

Reviewer 3 Report

  1. The NO.12 research of Jonathan J is conducted in mice, thus including it in this systematic review  is not suitable. It is not known whether the pathogenetic mechanism of ZIKV is the same in mice and human.
  2. In Table 1, the "Mousses"  in the line of NO.12 should be revised.
  3. In line 233, there is a full stop is missing between "death" and "In" .

Author Response

Dear Reviewer 3

Thank you very much for your review of my article!

  The NO.12 research of Jonathan J is conducted in mice, thus including it in this systematic review  is not suitable. It is not known whether the pathogenetic mechanism of ZIKV is the same in mice and human.

InTable 1, the "Mousses"  inthelineofNO.12 should be revised.(In both cases of research with mice, the virus was observed to have caused the same neurodegenerative lesions in the brains of the laboratory animals and the humans. Furthermore, mice are known to be a model organism with many similarities with the human one).

  In line 233, there is a full stop is missing between "death" and "In" .(Corrected line 269)

Reviewer 4 Report

Dear Authors,

I was very excited to read about the connection between Zika virus and the risk of developing microcephaly. I was hoping that with several years since the Zika virus pandemic, there would have been sufficient studies published to get a decent assessment of the connection. I was also looking forward to reading a good systematic literature review, and was even hoping to be able to use it as an example in one of my classes.

Unfortunately, I was disappointed when I started reading the paper. First, I noticed the disorganized nature of the Introduction, along with several issues with spelling and English language. Some of them could have been avoided just by proofreading the manuscript before submission (e.g. on line 47 as "measurevalue"). Some others would need someone who is a fluent English speaker to look through to correct in both spelling ("dangue" vs "dengue") as well as grammar.

One of the major criticism that I have is some major omissions in methodology. On the flow diagram on page 3, you go from 338 studies to 15 in one step, removing the rest of the studies for revisions (150) and 173 for other title subjects. There is not sufficient explanation why this was done and how, and why there were so many revisions and other title subjects. This was a major issue for me, because it could introduce bias into the study, depending on how you went from 338 to 15! I would need to be convinced that this is OK!

My other major concern included Table 2, which seem to describe the evaluation of the methodological quality of the studies included. This is in the Results, and there is no mention of this evaluation in the methods, so it is hard to understand what this evaluation is and how it works. 

There is no discussion about the ongoing risk of Zika virus transmission and microcephaly, which I'm really missing!

The last paragraph in the Conclusions talks about Greece, which I understand is the home country of the authors. I understand their interest, but Greece is not the focus of the study, and this discussion reads very much out of place. I would suggest to repeat the main conclusion of the study, but I'm not exactly sure what that is, which shows how unclear the study is.

Author Response

Dear Reviewer 4

Thank you very much for your recommendation

Unfortunately, I was disappointed when I started reading the paper. First, I noticed the disorganized nature of the Introduction, along with several issues with spelling and English language. Some of them could have been avoided just by proofreading the manuscript before submission (e.g. on line 47 as "measurevalue").( line 50-51)

Some others would need someone who is a fluent English speaker to look through to correct in both spelling ("dangue" vs "dengue") as well as grammar. (Corrected lines 69, 90)

One of the major criticism that I have is some major omissions in methodology. On the flow diagram on page 3, you go from 338 studies to 15 in one step, removing the rest of the studies for revisions (150) and 173 for other title subjects. There is not sufficient explanation why this was done and how, and why there were so many revisions and other title subjects. This was a major issue for me, because it could introduce bias into the study, depending on how you went from 338 to 15! I would need to be convinced that this is OK! (In this systematic review only cross-sectional studies and case reports were used. Therefore, reviews, systematic reviews and meta-analyses (total 150 studies) were rejected. 173 articles were rejected because their title was different from our subject; more specifically, apart from “zika virus and microcephaly” the key words included “zika virus and fetus toxicity”, “zika virus and craniostenosis” “zika virus and teratogenesis” “zika virus and children’s mental health”. As a result, a big number of articles not related to the subject of our own study, microcephaly was rejected. For example, studies including other genetic disorders related to zika, i.e. hearing problems, eye problems, cerebellum problems, hydrocephalus, fetal death, intrauterine growth restriction or even the Guillain-Barren syndrome).

My other major concern included Table 2, which seem to describe the evaluation of the methodological quality of the studies included. This is in the Results, and there is no mention of this evaluation in the methods, so it is hard to understand what this evaluation is and how it works. (Necessary corrections made,lines 161, 171-172)

There is no discussion about the ongoing risk of Zika virus transmission and microcephaly, which I'm really missing! (Added, lines 349-350. Reference to the prevention measures (CDC / MEDICHEM / WHO) is made in lines 314-336)

The last paragraph in the Conclusions talks about Greece, which I understand is the home country of the authors. I understand their interest, but Greece is not the focus of the study, and this discussion reads very much out of place. I would suggest to repeat the main conclusion of the study, but I'm not exactly sure what that is, which shows how unclear the study is. (Removed)

Round 2

Reviewer 2 Report

The paper is much improved but there are still issues that I feel should be addressed before it is ready for publication. Some of the issues are still grammatical and I have attempted to help in this regard while others are related to content. 

Line 34 - Eliminate "using all available research to reach a conclusion"

Line 35 - Rhesus monkeys

Line 46  -Insert The CDC (the same for the WHO wherever you refer to them) Suggestion to change sentence to: The CDC defines the Zika virus as a flavivirus, transmitted .... , and infection..... CDC

Line 52 - Suggestion for grammar - During routine ultrasounds in pregnancy, microcephaly can be diagnosed in the second or early third trimester.

Line 75 - What are primary mammals - do you mean primarily  or primate mammals

Line 80- Suggestion to rewrite as "After a 10 day incubation period, the mosquito's saliva becomes infected and  ......."

Line 86 - Change - "Apart to" to Apart from

Line 96 - ..... in contact with an infected person must be (not at) screened ( substitution for examined) with a serological blood test.

Line 99  Delete Also.  Exposed neonates should be evaluated ( suggested substitution)....

Line 105 - ...... the WHO

Line 112 - Change America into American travelers

Line 128 - Please insert a comma after found,

Line 133 - I still have an issue with your outcome as incidence of microcephaly as few if any of the studies included were able to determine incidence - frequency is more correct.

Line 136 - Please insert "them" after rate

Line 157- Flowchart - 150 articles were removed due to "revisions". Do you mean reviews as explained in your reply to my review. If not, then what do you mean by revisions.

Line 168- please spell out USA as United States of America (USA) the first time its mentioned 

Line 172 - "...and 3 were of moderate ....."

Line 177 - I understand that you want to show some statistical relevance by including the p values but by themselves they are meaningless unless you show what comparisons/ relationships they are representing.

Line 185 - Is this 15% of all the cases (8301) with microcephaly in the study. I suggest changing sample to patients or subjects with microcephaly.

Line 191 - You refer to all neonates were shown ..... but the preceding sentence only referred to 2 cases in the Melo study 2016 which does not have a reference. In Line 192 - who mention findings in the neonates including celiac disease! This is not diagnosed in neonates to my knowledge.       

Line 210 - I suggest you add the p-value here (although it is already in the table) or add a comment as to this not reaching statistical significance.

Line 211 - Change : "In some case reports" to ... "In the case report of Driggers et al, 2016 ... "as you only go on to describe this case report in the sentence. Alternatively you might consider starting this paragraph with "There were 3 case studies in our review". Then proceed. as this paragraph describes 3 case study reports.  

Line 225 - ...Change sentence as follows ; Also, in Martinez et al, 2016, samples from 2 neonates with microcephaly (estimated gestational ages of 36 and 38 weeks), dying within 20 hours of birth and 2 embryo miscarried at 11 and 13 weeks were sent to the CDC ......... for evaluation of ...... .  What did the CDC find? You need to show the relevance.

Line 230 - .".. but they had no clinical symptoms"

Line 232 - This sentence needs clarity. perhaps as follows: In the cross-sectional study of Pacheco et al 2016, data on 11.944 cases were collected from .....

Line 242 - Change fetus to fetuses

Line 267 -In the first case - change to "In the first model or instance.....

LIne 291 - Change "have" to "has"

Line 292 - Add "in" pregnancy

Line 300 - Change .. "Also important" to .... Also of importance  

Line 307 - Change "evade" to disregard

Line 338 - I recommend changing "ways" to "frequency" ... to reflect your study. Also ... as well as factors increasing the risk of vertical transmission in pregnancy. 

Line 345 - "prevent unintended pregnancies and take measures" (what measures e.g.  to combat transmission of Zika virus from infected mosquitoes especially ......... . 

Line 349 - "because the prevalence of Zika infection and risk of vertical transmission remains high'

Line 352 - Zika virus epidemic

Line 353  - 358- Suggestions: In addition to the huge financial losses due to a heavy dependence on tourism, and increased stress on the healthcare system, affected individuals with ....... .............. . The Zika epidemic has disproportionately affected ...... . As recently observed, epidemics spread by .... can expand rapidly, including to other parts of the world and further highlights the need for effective control of the vectors.  

Some of the above recommendations were not made in the first review but became more evident  in rereading your paper and the new sections were now reviewed for the first time. I do hope that with further work, this paper will be published.

Author Response

Dear reviewer

I would like to thank you for your valuable help in revising this article. Your corrections were important and useful. I quote you the changes in detail:

Line 34 - Eliminate "using all available research to reach a conclusion" (corrected)

Line 35 - Rhesus monkeys (corrected)

Line 46  -Insert The CDC (the same for the WHO wherever you refer to them) Suggestion to change sentence to: The CDC defines the Zika virus as a flavivirus, transmitted .... , and infection..... CDC (lines 46, 75, 105, 227, corrected)

Line 52 - Suggestion for grammar - During routine ultrasounds in pregnancy, microcephaly can be diagnosed in the second or early third trimester. (Line 52 corrected)

  Line 75 - What are primary mammals - do you mean primarily or primate mammals (I mean “primate”- corrected)

Line 80- Suggestion to rewrite as "After a 10 day incubation period, the mosquito's saliva becomes infected and  ......."(line 80 corrected)

Line 86 - Change - "Apart to" to Apart from (line 86 corrected)

Line 96 - ..... in contact with an infected person must be (not at) screened ( substitution for examined) with a serological blood test. (Lines 96-97 corrected)

Line 99  Delete Also.  Exposed neonates should be evaluated (suggested substitution).... (Line 99 corrected)

Line 105 - ...... the WHO (corrected)

Line 112 - Change America into American travelers (corrected)

Line 128 - Please insert a comma after found, (corrected)

Line 133 - I still have an issue with your outcome as incidence of microcephaly as few if any of the studies included were able to determine incidence - frequency is more correct. (Corrected)

Line 136 - Please insert "them" after rate (corrected)

Line 157- Flowchart - 150 articles were removed due to "revisions". Do you mean reviews as explained in your reply to my review. If not, then what do you mean by revisions. ( I mean reviews and I corrected  it.)

Line 168- please spell out USA as United States of America (USA) the first time its mentioned (corrected in line 74) 

Line 172 - "...and 3 were of moderate ....."(Line 174, corrected)

Line 177 - I understand that you want to show some statistical relevance by including the p values but by themselves they are meaningless unless you show what comparisons/ relationships they are representing. (Table 2 has been corrected)

Line 185 - Is this 15% of all the cases (8301) with microcephaly in the study. I suggest changing sample to patients or subjects with microcephaly. (Corrected)

Line 191 - You refer to all neonates were shown ..... (I mean both neonates, corrected) but the preceding sentence only referred to 2 cases in the Melo study 2016 which does not have a reference. (Line 194, reference 21). In Line 192 - who mention findings in the neonates including celiac disease! This is not diagnosed in neonates to my knowledge.  (I mean brain celiac disease and I completed it line 194)      

Line 210 - I suggest you add the p-value here (although it is already in the table) or add a comment as to this not reaching statistical significance. (Added)

 Line 211 - Change : "In some case reports" to ... "In the case report of Driggers et al, 2016 ... "as you only go on to describe this case report in the sentence. Alternatively you might consider starting this paragraph with "There were 3 case studies in our review". Then proceed. as this paragraph describes 3 case study reports. (Line 213, corrected) 

Line 225 - ...Change sentence as follows ; Also, in Martinez et al, 2016, samples from 2 neonates with microcephaly (estimated gestational ages of 36 and 38 weeks), dying within 20 hours of birth and 2 embryo miscarried at 11 and 13 weeks were sent to the CDC ......... for evaluation of ...... .  What did the CDC find? You need to show the relevance. (Lines 227-232, corrected)

Line 230 - .".. but they had no clinical symptoms"(Line 233, corrected)

Line 232 - This sentence needs clarity. perhaps as follows: In the cross-sectional study of Pacheco et al 2016, data on 11.944 cases were collected from ..... (Lines 235-237, corrected)

Line 242 - Change fetus to fetuses (Line 246 corrected)

Line 267 -In the first case - change to "In the first model or instance..... (Line 271, corrected)

LIne 291 - Change "have" to "has"(Line 295, corrected)

Line 292 - Add "in" pregnancy (Line 296, added)

Line 300 - Change .. "Also important" to .... Also of importance (Changed, line 304)

Line 307 - Change "evade" to disregard (Changed, line 311)

Line 338 - I recommend changing "ways" to "frequency" ... to reflect your study. Also ... as well as factors increasing the risk of vertical transmission in pregnancy. (Line 343-344, changed)

Line 345 - "prevent unintended pregnancies and take measures" (what measures e.g.  to combat transmission of Zika virus from infected mosquitoes especially ......... . (Line 351-352, added)

Line 349 - "because the prevalence of Zika infection and risk of vertical transmission remains high'(lines 355-356, corrected)

Line 352 Zika virus epidemic(Line 358, corrected)

Line 353  - 358- Suggestions: In addition to the huge financial losses due to a heavy dependence on tourism, and increased stress on the healthcare system, affected individuals with ....... .............. . The Zika epidemic has disproportionately affected ...... . As recently observed, epidemics spread by .... can expand rapidly, including to other parts of the world and further highlights the need for effective control of the vectors.  (Lines 358-365, corrected)

Yours Sincerely

Evangelia Antoniou

Reviewer 4 Report

Dear Authors,

I very much appreciate the extensive changes that you did in your revision. The paper reads much better. However, I still had a number of stylistical corrections and comments. Since you submitted the Word document of the paper, I added my corrections and comments into your document, with Track Changes so you can see them. I also read your Author's response, and realized you did not include some of the details on how many papers were excluded from the study and why into the main manuscript text. Please do that. There is also some clarification required in the criteria that you used to evaluate the selected papers. I tried to guess some of them, but please double-check so I'm not misinterpreting what you did. At this point, I do consider your paper to be much more clear, and it provides a nice overview of the evidence we have for the connection between Zika virus infection and microcephaly.

Krisztian Magori

Author Response

Dear Krisztian Magori, 

First of all, I would like to thank you for your recommendations that helped the paper a lot. I quote below the further changes that have been made:

I also read your Author's response, and realized you did not include some of the details on how many papers were excluded from the study and why into the main manuscript text. Please do that. (Information has been added, lines 130-136)

There is also some clarification required in the criteria that you used to evaluate the selected papers. I tried to guess some of them, but please double-check so I'm not misinterpreting what you did. At this point, I do consider your paper to be much more clear, and it provides a nice overview of the evidence we have for the connection between Zika virus infection and microcephaly. (The methodological quality criteria have been clarified, lines 144-155)

Yours Sincerely

Evangelia Antoniou